# Sirolimus-Embedded Silk Microneedle Wrap to Prevent Neointimal Hyperplasia in Vein Graft Model

**DOI:** 10.3390/ijms24043306

**Published:** 2023-02-07

**Authors:** Jung-Hwan Kim, Eui Hwa Jang, Ji-Yeon Ryu, Jiyong Lee, Jae Ho Kim, Wonhyoung Ryu, Young-Nam Youn

**Affiliations:** 1Division of Cardiovascular Surgery, Department of Thoracic and Cardiovascular Surgery, Severance Cardiovascular Hospital, Yonsei University College of Medicine, Seoul 03722, Republic of Korea; 2School of Mechanical Engineering, Yonsei University, Seoul 03722, Republic of Korea

**Keywords:** neointimal hyperplasia, vein graft, microneedle, silk fibroin, perivascular drug delivery, vascular wraps

## Abstract

We investigated the role of a sirolimus-embedded silk microneedle (MN) wrap as an external vascular device for drug delivery efficacy, inhibition of neointimal hyperplasia, and vascular remodeling. Using dogs, a vein graft model was developed to interpose the carotid or femoral artery with the jugular or femoral vein. The control group contained four dogs with only interposed grafts; the intervention group contained four dogs with vein grafts in which sirolimus-embedded silk-MN wraps were applied. After 12-weeks post-implantation, 15 vein grafts in each group were explanted and analyzed. Vein grafts applied with the rhodamine B–embedded silk-MN wrap showed far higher fluorescent signals than those without the wrap. The diameter of vein grafts in the intervention group decreased or remained stable without dilatation; however, it increased in the control group. The intervention group had femoral vein grafts with a significantly lower mean neointima-to-media ratio, and had vein grafts with an intima layer showing a significantly lower collagen density ratio than the control group. In conclusion, sirolimus-embedded silk-MN wrap in a vein graft model successfully delivered the drug to the intimal layer of the vein grafts. It prevented vein graft dilatation, avoiding shear stress and decreasing wall tension, and it inhibited neointimal hyperplasia.

## 1. Introduction

Bypass graft surgery is the treatment of choice for extensive arterial occlusive disease that is not indicated for percutaneous intervention, or when interventional procedures have failed [1,2]. Additionally, autologous vein grafts are widely used for small-sized arterial disease, such as coronary artery or peripheral artery occlusive disease, because prosthetic vascular grafts have shown a high occlusion rate in small-sized arteries [3,4]. Despite the superiority of autologous vein grafts over prosthetic grafts in terms of patency, the long-term vein graft failure rate was reported to be 40~50% in coronary artery bypass grafts [5,6], leading to worse clinical outcomes in terms of death, myocardial infarction, or repeat revascularization [7].

The pathophysiology of vein graft failure is known to involve several distinct phases. Graft failure in the early phase (<1 month after grafting) is characterized by thrombosis, which is closely related to the technical or anatomical problem. Additionally, late failure is characterized by neointimal hyperplasia (within 1–24 months after grafting), eventually transitioning to atherosclerosis of the vein graft (>24 months after grafting) [8,9]. Several pharmacologic therapies including dual antiplatelet therapy and lipid-lowering statins have been emphasized by current revascularization guidelines as important strategies in the prevention of vein graft failure [10].

In recent studies, different forms of external vascular devices such as films, wraps, depot gels, meshes, rings, or microparticles/nanoparticles were reported for perivascular delivery of an antiproliferation drug such as paclitaxel or sirolimus to prevent neointimal hyperplasia [11,12,13,14,15]. In our previous studies, an external applicable perivascular microneedle (MN) device with paclitaxel and sirolimus showed appropriate drug distribution in vascular tissue and significantly decreased neointimal hyperplasia of the abdominal aorta in a rabbit balloon injury model [16,17]. Because silk showed excellent cell compatibility [18,19] and the potential for maintained drug delivery with the well-controlled release of drugs [20,21], we used a highly flexible and porous silk fibroin MN wrap for localized perivascular drug delivery in a rabbit balloon injury model [22]. Our results showed enhanced cell compatibility and successful delivery of the antiproliferative drug. Moreover, the vascular tissue maintained its biological and structural unity with minimal deformation even after the long-term wrapping. Silk MNs successfully penetrated the target vascular tissue to the intima layer and successfully released the antiproliferation drug. As a result, silk MN wraps effectively inhibited neo-intimal hyperplasia with 62.1% reduced neointimal formation, 39.1% decreased neointima-to-media ratio, and 71.8% inhibited smooth muscle cell proliferation in the neointimal region compared with bare silk MN wraps.

Few studies have investigated the effects of a perivascular drug delivery device on neointimal hyperplasia or vascular remodeling in large animal vein graft models. Based on our previous studies in the rabbit arterial injury model, a more complex and clinically applicable experiment was designed. In this study, we developed a large animal vein graft model using peripheral arteries and veins from dogs and investigated the role of drug-embedded silk MN wrap as an external vascular device for drug delivery efficacy, the influence of physical properties, the inhibition of neointimal hyperplasia, and vascular remodeling.

## 2. Results

### 2.1. Drug Release of Silk MNs

To verify the appropriate drug release to the intima layer of the vein graft from the silk MN, a fluorescence analysis was used with the RhoB-embedded silk MN wrap. As shown in Figure 1A, ITA-vein grafts applied with the RhoB-embedded silk MN wrap showed far higher fluorescent signals than those without wrap. Because sirolimus is an mTOR inhibitor, we identified whether mTOR activation was effectively inhibited when applying the sirolimus-embedded silk MN wrap to the ITA-vein graft. Figure 1B shows that mTOR expression tended to decrease in the intervention group compared with preimplantation and the control group; however, this difference was not significant.

Additionally, to verify the appropriate sirolimus release to the vein graft from the silk MN, we used LC-MS analysis. As shown in Figure 2, sirolimus and ascomycin were detected at 10.7 and 11.5 min using LC-MS analysis. The calibration curve of sirolimus was y = 0.226683× + 0.396562 (R2 = 0.999) and showed appropriated linearity in a range of 0.05–50.00 µg/mL. In the three ITA-vein grafts, the sirolimus concentration was 15.54 µg/mL, 22.03 µg/mL, and 95.68 µg/mL, respectively, with a mean concentration of 44.42 µg/mL.

### 2.2. Diameter and Velocity Changes of Vein Grafts

The diameter of vein grafts in the control group tended to increase in femoral vein grafts and significantly increased in jugular vein grafts. However, in the intervention group, the diameter decreased significantly in femoral vein grafts and remained stable without dilatation in jugular vein grafts compared with the control group (Figure 3A). Additionally, the velocity of vein grafts was significantly increased immediately after surgery in both groups, but it tended to decrease during the follow-up period, decreasing to a level similar to before surgery in the control group. In the intervention group, the velocity of vein grafts was not significantly decreased and remained stable, especially in jugular vein grafts (Figure 3B).

### 2.3. Neointimal Hyperplasia in Vein Graft

We assessed the efficacy of implanted sirolimus-embedded silk MN wrap at 12 weeks after surgery. First, both H&E-stained histological images were examined to visualize the thickening of the intima after implantation (Figure 4). Next, PCNA and alpha-smooth muscle actin (α-SMA) expression levels were measured as indicators of smooth muscle cell proliferation and differentiation, respectively (Figure 5).

According to the H&E staining results, the intervention group had femoral vein grafts with a significantly lower mean neointima-to-media ratio (37.2 ± 20.54) than the control group (103.42 ± 53.51, *p* = 0.01). For jugular vein grafts, the means neointima-to-media ratio was lower in the intervention group (54.35 ± 35.19) than in the control group (78.59 ± 50.19), but this difference was not statistically different (*p* = 0.29) (Figure 4).

PCNA expression levels of implanted vein grafts were not significantly different at 12 weeks in both groups (Figure 5A). For femoral vein grafts, α-SMA expression levels were significantly lower in the intervention group compared to the control group. However, the α-SMA expression levels of jugular vein grafts were not significantly different between groups (Figure 5B).

### 2.4. Neointimal Hyperplasia in Vein Graft

According to the Masson’s trichrome staining results, the femoral vein graft’s intima layer had a significantly lower collagen density ratio in the intervention group (125.22 ± 8.75) than in the control group (136.92 ± 4.49; *p* = 0.03). Additionally, the jugular vein graft’s intima layer had a significantly lower collagen density ratio in the intervention group (120.08 ± 13.55) than in the control group (144.82 ± 10.15; *p* < 0.01) (Figure 6).

We examined autophagy by the expression of LC3B, which is widely used as a biomarker of autophagy, and LC3B-II/LC3B-I expression was significantly higher in the intervention group than in the control group (*p* = 0.05; Figure 7). Additionally, apoptotic cell counts shown on TUNEL staining at 12 weeks after surgery were significantly higher in the intervention group (790.86 ± 52.34 cells) than in the control group (1144.13 ± 258.17 cells; *p* < 0.01; Figure 8).

## 3. Discussion

Despite several theoretical advantages [11,12,13,14,15], a perivascular drug delivery device should resolve several challenges in terms of inflammatory change to the treated tissue after degradation, the influence of vascular constriction, controlled drug release, ease of application, or drug localization. In our previous study [22], a highly flexible and porous silk fibroin MN wrap was shown to enhance cell compatibility and maintain the biological and structural unity of vascular tissue with minimal deformation. Additionally, it successfully penetrated the target vascular tissue to the intima layer and successfully released the antiproliferation drug.

In this study, a sirolimus-embedded silk MN wrap was successfully delivered to the intimal layer of vein grafts and released to the vascular tissue in a large animal vein graft model. The antiproliferative mechanism of sirolimus is inhibition of the mTOR-mediated signal-transduction pathways [23]. We noticed a decreased level of mTOR expression compared with the preintervention and the control group, although the decrease was not statistically significant. An LC-MS analysis of sirolimus in vein grafts showed that MNs successfully penetrated vein graft tissue and sirolimus was released to the intimal layer on vein grafts.

The remodeling processes of the vein graft occur within several days after grafting, and it leads to the neointimal hyperplasia within several months that is the primary cause of intermediate phase vein graft failure [8,9]. Endothelial cell damage causes the expression of various growth factors and cytokines, which promotes excessive extracellular matrix deposition in the neointimal layer and smooth muscle cells (SMCs) proliferation and migration toward the neointimal layer of the vein graft [23]. Extracellular matrix deposition and the proliferation, migration, and death of SMCs are important components in neointimal hyperplasia formation. During migration, SMCs change from a quiescent contractile phenotype to a dedifferentiated, proliferating, or synthetic phenotype [24,25,26].

Alpha-smooth muscle actin, encoded by the ACTA2 gene, typically expresses in the vascular SMCs and contributes to vascular motility and contraction [27]. Studies of α-SMA and ACTA2 mutations demonstrated that it caused various vasculopathies such as the early onset of coronary artery disease, ischemic strokes, and familial thoracic aortic aneurysms and dissections [28]. Proliferative and secretory activities of α-SMA, as well as transition from a contractile to a dedifferentiated, proliferating, or synthetic phenotype of α-SMA, were also considered to be the underlying mechanism of vasculopathies [29]. A recent vein graft rat animal study, where the jugular vein was grafted to carotid artery, showed that α-SMA expression was significantly decreased in vein grafts in which intimal hyperplasia was more suppressed. This study showed that the local application of sirolimus contributed to inhibiting intimal hyperplasia the and α-SMA expression of the vein graft [30].

Our study showed that silk MN wrap prevented vein graft dilatation and maintained the flow velocity. The vein graft interposed to the arterial system is exposed to a high pressure and flow of the arterial environment and it results in morphological and histological changes to the vein graft [31]. A vein is required to increase in dimension to adapt to the high pressure in the arterial system [32]. This initial adaptation process can lead to vein graft failure. Intimal hyperplasia and vascular wall thickening are caused by SMCs proliferation and extracellular matrix deposition induced by increased pressure, shear stress, and inflammatory responses [33]. Several studies in human tissue cultures and experimental models showed that the use of an external device could decrease shear stress and the wall tension of the vein graft, thereby inhibiting SMCs proliferation, negative remodeling, and neointimal hyperplasia [34,35,36,37,38]. Theoretically, external stenting reduces wall tension and the stretching of endothelial cells, and it plays a role as a protective outer layer [39]. Taggart et al. showed that external stenting to the vein graft reduced intimal hyperplasia after coronary artery bypass surgery [39], and Ferrari et al. reported that an external mesh device could improve long-term graft durability [40].

The results of our study revealed differences in vein graft diameter, velocity changes, and neointimal formation. The femoral vein graft diameter and velocity decreased with the passage of time after surgery, whereas the jugular vein graft remained stable without significant changes. Additionally, the mean neointima-to-media ratio of jugular vein grafts was not statistically different between the control and intervention groups. The differences in results between femoral and jugular vein grafts are thought to be due to hemodynamic disparity and the amount of media (SMC). The carotid artery originates directly from the aortic arch, but the femoral artery originates from several major branches such as supra-aortic vessels, visceral arteries, intercostal arteries, and the internal iliac artery. Thus, long-standing pressure and shear stress are higher in the carotid artery than the femoral artery. Moreover, it is known that the carotid artery is composed of larger SMC-rich media compared with the femoral artery [41]. These hemodynamic disparities and the amount of SMC layer are important factors that make the femoral vein graft more affected by the antiproliferative effect of sirolimus.

Atherosclerotic lesion formation is characteristic of late phase vein graft failure [8,9]. If vein graft wall inflammation fails to resolve, then the remodeling process is toward pathologic remodeling, fibrosis, and vein graft stenosis [42]. Apoptosis is known to be related with various characteristics of fibrosis as initiators or perpetuators of the fibrotic response [43]. However, the role of apoptosis in vascular remodeling remains unclear. Several studies on antihypertensive agents, such as calcium channel blockers [44] and angiotensin-converting enzyme inhibitors [45], have suggested that apoptotic mechanisms might contribute to the regression of vascular wall growth. Several studies reported the beneficial role of autophagy in atherosclerosis. Kiffin et al. showed that autophagy of the damaged components could protect plaque cells against oxidative stress and could promote cell survival [46]. In addition Martinet et al. investigated the protective role of autophagy in atherosclerosis with an in vitro experiment [47]. It showed that SMC death induced by low concentrations of statins was attenuated by the autophagy inducer 7-ketocholesterol. Schrijver et al. explained that the inhibition of autophagy was harmful because this would accelerate other forms of cell death such as necrosis, and impair the phagocytic process [48]. Our study also showed that vein grafts with sirolimus-embedded silk MN wrap have a high level of apoptotic cells, autophagy activity, and a low level of collagen density.

This study had several limitations. It had a relatively small sample size, so some results did not show statistical significance, even though there was a tendency for differences. To validate the mechanical effect of silk MN wrap, a better design would be to divide the sample into three groups: control, silk MN wrap without sirolimus, and silk MN wrap with sirolimus. We performed an unpublished preliminary study using bare silk MN wrap without drugs to the rabbit abdominal aorta injury model. This study showed that bare silk MN wrap was not associated with the prevention of neointimal hyperplasia (Appendix A). Thus, we concluded that silk MN wrap itself without a drug could not prevent neointimal hyperplasia, and we did not include the bare silk MN wrap in this study. The double staining of PCNA and α-SMA should be very helpful interpreting the results, but our study did not include these data. Finally, it would be important to identify the systemic effect of sirolimus-embedded silk MN wrap through serum sirolimus levels, but we could not do this.

## 4. Materials and Methods

### 4.1. Materials

Sirolimus (rapamycin, R-5000) was purchased from LC Laboratories (Woburn, MA, USA). Polydimethylsiloxane (PDMS; Sylgard^®^ 184 silicone elastomer kit) was purchased from Dow Corning (Midland, MI, USA). Rhodamine B (RhoB, R6626) and Dulbecco’s phosphate-buffered saline (D8662) were purchased from Sigma Aldrich (St. Louis, MO, USA). Dimethyl sulfoxide (D0457) was purchased from Samchun Chemical (Seoul, Korea). Zoletil^®^ 50 was purchased from VIRBAC (Virbac, Carros, France), and heparin sodium was obtained from Hanlim (Seoul, Korea). Aspirin was purchased from BayerKorea (Seoul, Korea), and clopidogrel bisulfate was purchased from Plavix (Sanofi Winthrop Industries, Paris, France). Solvents for high-performance liquid chromatography were purchased from Merck Inc. (Darmstadt, Germany). All chemicals and reagents were of analytical grade. Radioimmunoprecipitation assay (RIPA) lysis buffer (R0146CD) was obtained from Bylabs (Hanam, Korea). mTOR (#2972) and phospho-mTOR (Ser2248, #2971) were purchased from Cell Signaling Technology (Beverly, MA, USA). Proliferation cell nuclear antigen (PCNA) (ab19166), α-SMA (ab5694), LC3B (ab48394), and β-actin (mAbcam 8224) were purchased from Abcam (Cambridge, MA, USA).

### 4.2. Fabrication of Silk Wrap and Sirolimus-Embedded Silk MN Wrap

Fabrication of a silk MN and embedment of sirolimus to the MN was described in our previous study [16,49]. Briefly, a porous silk wrap was fabricated using a freeze-drying method. An aqueous silk solution was poured into the petri dish, frozen for 1 h for making silk plates, and the frozen silk plate was lyophilized. After 24 h, the porous silk wrap was cut and treated in 100% EtOH for 1 min for crystallization. Finally, after washing in deionized water several times and after 24 h of drying, the highly porous silk wrap (10 mm width; 400 μm thickness) was fabricated.

The transfer molding method was used to compose the sirolimus-embedded silk MNs on the porous silk wrap. Negative PDMS molds, which contained MNs shapes, were prepared using sirolimus-MN masters. Sirolimus-embedded silk MNs were prepared to have 1 μg of the drug for each MN device, as in our previous studies [49]. Next, the sirolimus-silk formulation was poured on the negative MN molds and centrifuged for 5 min, then dried for 30 min. After one more repeated step and overnight drying, the silk solution was transferred to a pre-molded silk MN structure using a micropillar for making silk bumps as adhesion layers according to the following steps: (1) align the micropillar with the transferred silk solution to the pre-molded silk MN; (2) make contact between the solution and MN; (3) draw the micropillar and form the capillary bridge; (4) create the silk MN array with bumps for the adhesion layer; and (5) form silk MNs on the lyophilized silk wrap. Finally, the silk MN wrap, which had sirolimus-embedded silk MNs on a highly flexible and porous silk wrap, was fabricated (Figure 9).

### 4.3. Animals and In Vivo Surgical Procedures

The experimental procedure for all in vivo animal studies and animal care protocol (IAUCU no. 2018–0207) was approved by the Institutional Animal Care and Use Committee of Yonsei University Health System (Severance Hospital, Korea). For all animal studies using mongrel dogs, experimental procedures were carried out in accordance with the Guide for the Care and Use of Laboratory Animals (National Research Council, Washington, DC, USA). The mongrel dogs were divided into two groups: control (*n* = 4) and intervention (*n* = 4). The experimental procedures were described in our previous study [49].

General endotracheal anesthesia was induced for all dogs with Zoletil (10 mg/kg) and xylazine (5 mg/kg). In the supine position, an anterior medial neck incision parallel to the trachea and a bilateral vertical groin incision were made. The bilateral common carotid artery, external jugular vein, common femoral artery, and common femoral vein were then exposed and dissected. After heparin sodium (80 U/kg) administration, the bilateral external jugular vein grafts were harvested with a length of 3 cm and interposed to each side of common carotid artery with end-to-end fashion using 6–0 polypropylene continuous sutures. Next, the bilateral common femoral vein grafts were interposed to each side of the common femoral artery with same manner. Arterial clamp times were 15 min or less in all cases. After reperfusion and bleeding control, the surgical procedure was completed for the control group. The sirolimus-embedded silk MN wrap was carefully wrapped around the vein graft to include both anastomosis sites and subsequently fixed with a surgical clip (LIGACLIP^®^, titanium, medium, Mexico) for the intervention group (Figure 10). Figure 11 showed detailed surgical procedures in the intervention group. The incisions were closed, and the animals were allowed to recover. Aspirin (100 mg/day) and clopidogrel bisulfate (75 mg/day) were administered from the day of surgery and maintained during that follow-up period. The implanted vein grafts were extracted and analyzed at 12 weeks after interposition, and the animal was sacrificed under general anesthesia. During the follow-up period, two femoral vein grafts (one graft in each group) were occluded and excluded in this study. In total, 15 vein grafts (eight jugular vein and seven femoral vein) in each control and intervention group were analyzed in this study.

### 4.4. In Vivo Drug Release Profile of Silk MNs

Bilateral internal thoracic arteries (ITA) were harvested from two dogs in the intervention group to identify drug release. The ITA harvesting was performed immediately after the explantation of both femoral and jugular vein grafts; the dogs were still alive at 12 weeks after the initial procedure. After median sternotomy, bilateral ITAs were harvested, then the femoral vein graft was anastomosed to ITA with end-to-end fashioning. Additionally, the RhoB- or sirolimus-embedded silk MN wrap was wrapped around the vein graft including both anastomosis sites and fixed with a surgical clip. The implanted vein grafts were extracted after 2 h from anastomosis and analyzed. One ITA-vein graft with RhoB-embedded silk MN wrap and three ITA-vein grafts with sirolimus-embedded silk MN wrap were analyzed for drug release. The animals were sacrificed under general anesthesia after all procedures were completed.

A fluorescence analysis was used for drug delivery efficiency analysis of the silk MN wrap in the same manner as our previous study [3]. Briefly, approximately 1 µg Rho B was embedded to silk MN and wrapped to a ITA-vein graft following the previous method. The harvested ITA-vein graft was embedded in an optimal cutting temperature compound following a cryosection protocol. The sample was then transversely sectioned in 10-μm thickness and contained at −20 °C. The cross-sectioned sample was evaluated with the constant exposure time at 0.2 s for fluorescent analysis.

The liquid chromatography–mass spectrometry (LC-MS) analysis was used to identify the sirolimus release from the silk MN wrap. The sirolimus-embedded silk MN wrap was wrapped to three ITA-vein grafts, and vein grafts were harvested following the previous method. Sirolimus from vein grafts was quantified using a 1290 Infinity liquid chromatography–6530 Accurate-Mass Q-TOF mass spectrometer (Agilent Technologies, Santa Clara, CA, USA) equipped with a Kinetex^®^ 2.6 μm EVO C18 100 Å LC column (100 × 2.1 mm, Phenomenex, Torrance, CA, USA), a quaternary pump, and an auto sampler. Separation was performed using 0.1% (*v*/*v*) formic acid, 6.5 mM ammonium bicarbonate in distilled water (A) and 0.1% (*v*/*v*) formic acid, and 6.5 mM ammonium bicarbonate acetonitrile (B) as eluents at a flow rate of 0.35 mL/min with the following gradient: 0–2 min, 5% B; 2–12 min, 100% B; 12–15 min, 100% B; 15–15.5 min, 5% B; and 15.5–17 min, 5% B. The injection volume was 5 μL, and the oven temperature was set at 45 °C. The SRM transitions of m/z 931.6 → 912.6 and m/z 928.6 → 790.5 were applied for sirolimus and ascomycin (internal standard), respectively. Mass data and data analysis were undertaken using Agilent MassHunter Workstation LC/MS Data Acquisition software for 6200 series TOF/6500 series Q-TOF (B.05.01) and processed by Agilent MassHunter Qualitative Analysis software (B.06.00, Agilent Technologies, Santa Clara, CA, USA).

### 4.5. Doppler Ultrasonography

The ultrasonographic procedures were described in our previous study [49]. All experimental animals were subjected to image analysis under general anesthesia and maintained at a mean blood pressure of 60 ± 10 mmHg. Doppler ultrasonography (S8Exp, SonoScape, China) was used to characterize the luminal diameter and pulse wave velocity of the vein graft before and 2, 4, 8, and 12 weeks after surgery. The proximal and distal portions of the femoral artery and a middle portion of the vein graft was imaged and measured from a defined portion of the longitudinal view with an ultrasound incident angle less than 60°. The degree of dilatation and velocity of the implanted vein graft after surgery were analyzed by calculating the change using the preoperational diameter and velocity as a reference point (Figure 10B).

### 4.6. Histopathological Analysis

The procedures of histopathological analysis were described in our previous study [49]. Histological examination of the implanted vein grafts was performed before and 12 weeks after surgery. The harvested samples were fixed in 10% buffered formalin, and cross-sections (5 µm thick) were cut and embedded into paraffin. The morphology of the vessel wall thickness and intimal hyperplasia was analyzed to clarify the layer division using hematoxylin and eosin (H&E) staining. The structure of each histological sections was manually identified using light microscopy (OLYMPUS, BX53). The lumen area, neointimal layer thickness, and medial layer thickness were measured using Image J software version 1.52v (National Institutes of Health, Bethesda, Maryland, USA). The neointima-to-media ratio was assessed. All pathological analysis was performed in a blinded manner.

The collagen density was analyzed using Masson’s trichrome staining. The stains were examined using light microscopy and measurements were made according to the intensity of the color blue, which represented the collagen density. A quantitative analysis was obtained by using the ratio of collagen density in the total area.

Apoptosis was detected by the terminal deoxynucleotidyl transferase dUTP nick-end labeling (TUNEL) method using an in situ apoptosis detection kit (TREVIGEN 4810-30-k). The number of apoptotic cells was counted at 200× magnification in an average of four sections per stained slide using Image J software.

### 4.7. Western Blot Analysis

Tissue samples were homogenized with an RIPA lysis buffer (Bylabs, R0146CD) containing EDTA-free protease inhibitor cocktail and centrifuged at 13,000 rpm for 10 min at 4℃. The supernatants were collected, and total protein content was measured using the Pierce BCA protein assay kit protocol (Thermo Scientific, Ref. 23227). Aliquots of the lysates were separated with 7.5–15% sodium dodecyl sulfate-polyacrylamide gel electrophoresis and transferred to polyvinylidene fluoride membranes (Bio-RAD, HC, USA). After blocking with 1% bovine serum albumin (BSA), the membrane was incubated for 24 h with primary antibodies, followed by 1 h with secondary antibodies in 1% BSA. Most primary antibodies were used at 1:1000 dilutions except β-actin (1:10,000), and a horseradish peroxidase (HRP)-conjugated goat anti-rabbit IgG (H+L) and HRP-conjugated goat anti-mouse IgG were used as secondary antibodies at a dilution of 1:5000. Protein bands were detected using the West-Q Pico Dura ECL Solution (GenDEPOT, Houston, TX, USA).

### 4.8. Statistical Analysis

All data were expressed as the means ± standard deviation. A Shapiro-Wilk test was used for the normality test, and normally distributed continuous variables were compared using a Student’s *t*-test and a one-way analysis of variance. In all tests, statistical significance was defined as a *p*-value less than 0.05. Statistical analyses were performed using SPSS for Windows, release 25.0 (IBM Corp., Armonk, NY, USA).

## 5. Conclusions

In this study, we proposed a sirolimus-embedded silk MN wrap to prevent neointimal hyperplasia in a large animal vein graft model. It successfully delivered the drug to the intimal layer of the vein graft and released it to vascular tissue. Additionally, this intervention affected the vein graft remodeling process. Silk MN wrap prevented vein graft dilatation and maintained flow velocity, preventing shear stress and decreasing wall tension. Moreover, the sirolimus-embedded silk MN wrap inhibited neointimal hyperplasia, represented by a decreased mean neointima-to-media ratio and a lower level of α-SMA expression. As a result, the remodeling process shifted toward inhibiting fibrosis by enhancing cell apoptosis and autophagy activity.

## Figures and Tables

**Figure 1 ijms-24-03306-f001:**
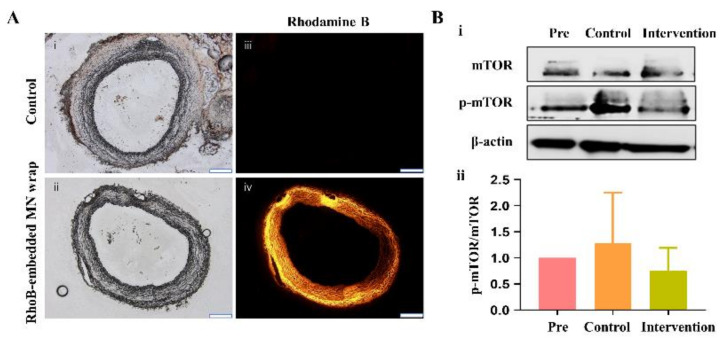
Drug release efficacy using fluorescence analysis and western blot analysis of mTOR expression. (**A**) Fluorescence analysis of internal thoracic artery-vein graft applied with rhodamine B (RhoB)–embedded silk microneedle (MN) wrap; differential interference contrast (DIC) microscopic images of a (i) control vein graft or (ii) vein graft wrapped with RhoB-embedded MN wrap; fluorescent images of a (iii) control vein graft or (iv) vein graft wrapped with RhoB-embedded MN wrap. (**B**) (i) Western blot analysis of mTOR expression before sirolimus-embedded silk MN wrap in the control and intervention groups and (ii) quantitative analysis of mTOR expression before sirolimus-embedded silk MN wrap in the control and intervention groups. Scale bars indicate 200 μm.

**Figure 2 ijms-24-03306-f002:**
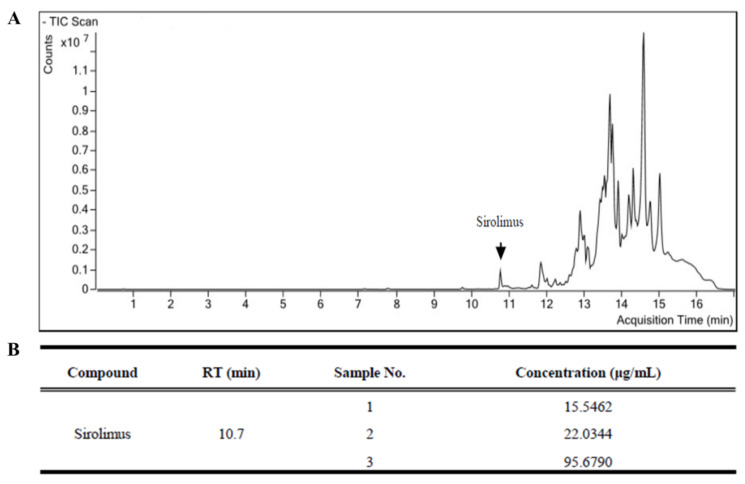
The liquid chromatography–mass spectrometry (LC-MS) analysis identifying the sirolimus release from silk microneedle (MN) wrap. (**A**) Appropriate detection of sirolimus at 10.7 min and (**B**) concentration of sirolimus in three vein grafts wrapped with sirolimus-embedded silk MN wrap.

**Figure 3 ijms-24-03306-f003:**
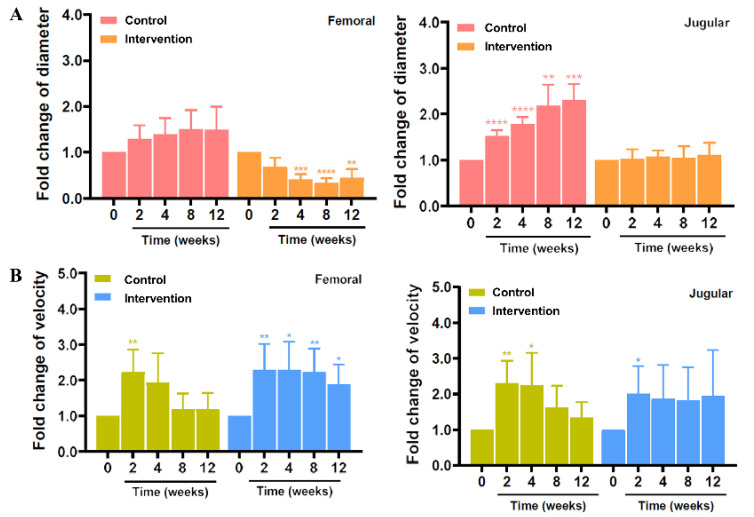
Mechanical effect of silk MN wrap to the vein graft. The relative fold change of (**A**) diameter and (**B**) velocity of femoral and jugular vein grafts measured by Doppler ultrasonography in the control and intervention groups (* *p* < 0.05; ** *p* < 0.01; *** *p* < 0.001; **** *p* < 0.0001).

**Figure 4 ijms-24-03306-f004:**
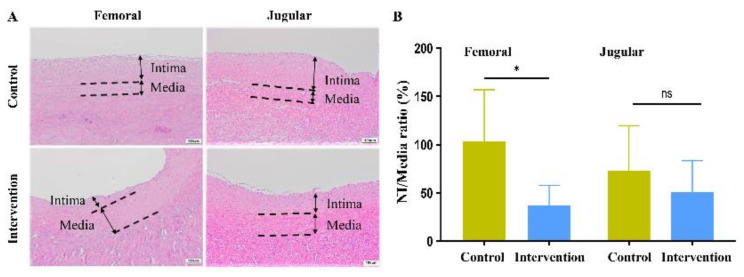
Inhibition of vein graft’s neointimal hyperplasia by sirolimus-embedded silk MN wrap. (**A**) Histological images with hematoxylin and eosin staining to visualize the thickening of the intima and media after sirolimus-embedded silk microneedle wrap implantation. (**B**) The neointima-to-media ratio of femoral and jugular vein grafts in the control and intervention groups (* *p* < 0.05; ns, *p* > 0.05). Scale bars indicate 100 μm.

**Figure 5 ijms-24-03306-f005:**
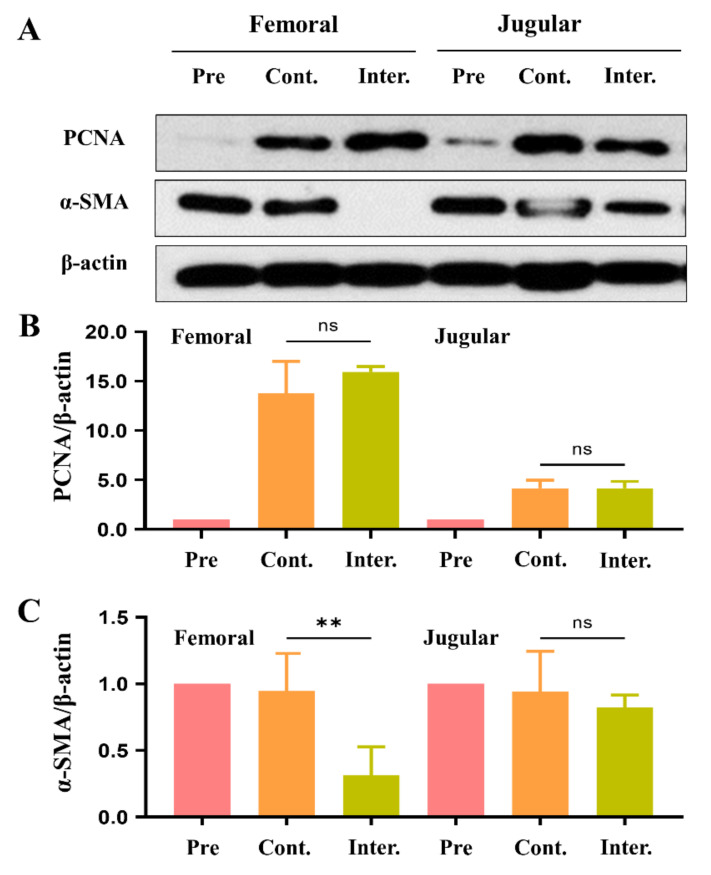
PCNA and α-SMA expression related to smooth muscle cell proliferation and differentiation. (**A**) Western blot analysis of PCNA, α-SMA, and β-actin and (**B**,**C**) quantitative analysis of (**B**) PCNA and (**C**) α-SMA expression of implanted vein grafts at preimplantation and in control and intervention groups (** *p* < 0.01; ns, *p* > 0.05).

**Figure 6 ijms-24-03306-f006:**
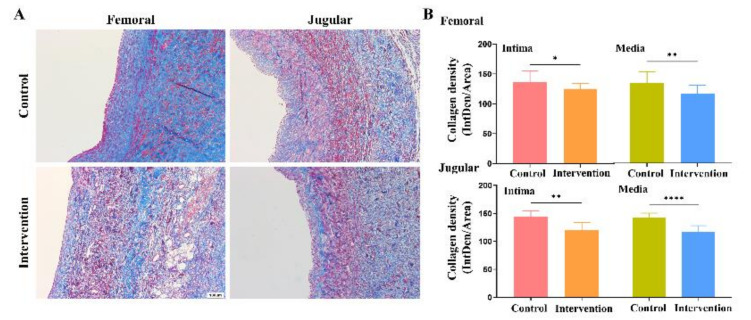
Collagen density of vein graft. (**A**) The collagen density (blue color) analysis with Masson’s trichrome staining and (**B**) quantitative analysis using the ratio of collagen density in the total area (* *p* < 0.05; ** *p* < 0.01; **** *p* < 0.0001). Scale bars indicate 100 μm.

**Figure 7 ijms-24-03306-f007:**
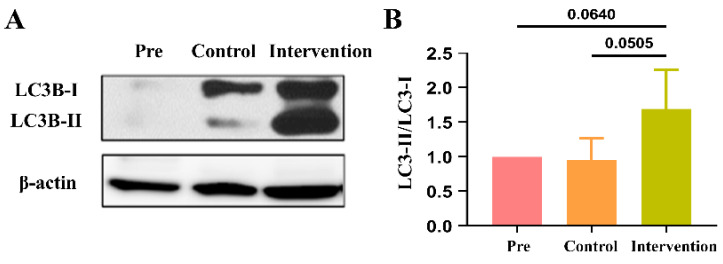
LC3B expression related to cell autophagy. (**A**) Western blot analysis of LC3B expression and (**B**) quantitative analysis of LC3B-II/LC3B-I expression of implanted vein grafts.

**Figure 8 ijms-24-03306-f008:**
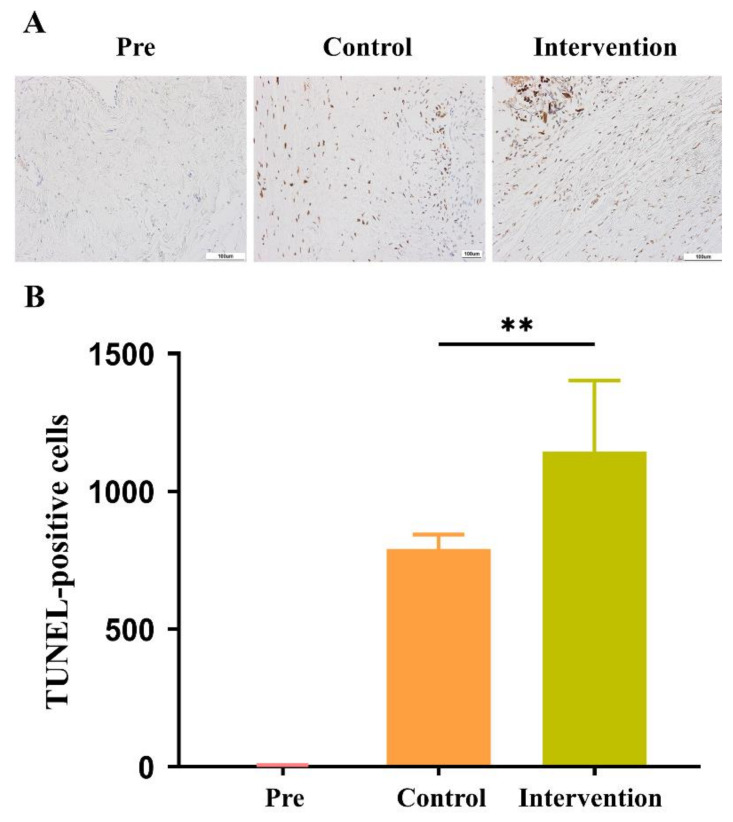
Cell apoptosis examined with terminal deoxynucleotidyl transferase dUTP nick-end labeling (TUNEL) staining. (**A**) Apoptotic cells (brown) shown after TUNEL staining and (**B**) the number of apoptotic cells counted at 200× magnification in an average of four sections per stained slide using Image J (** *p* < 0.01). Scale bars indicate 100 μm.

**Figure 9 ijms-24-03306-f009:**
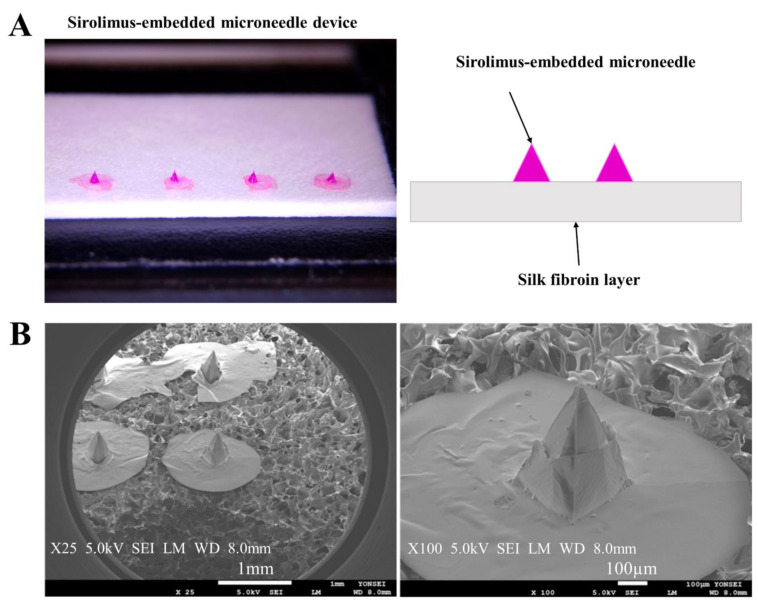
Design of sirolimus-embedded silk microneedle (MN) wrap. (**A**) Schematic design of the sirolimus-embedded silk MN wrap and (**B**) scanning electron microscope images of silk MN arrays on the porous silk wrap. Scale bars indicate 1 mm and 100 μm.

**Figure 10 ijms-24-03306-f010:**
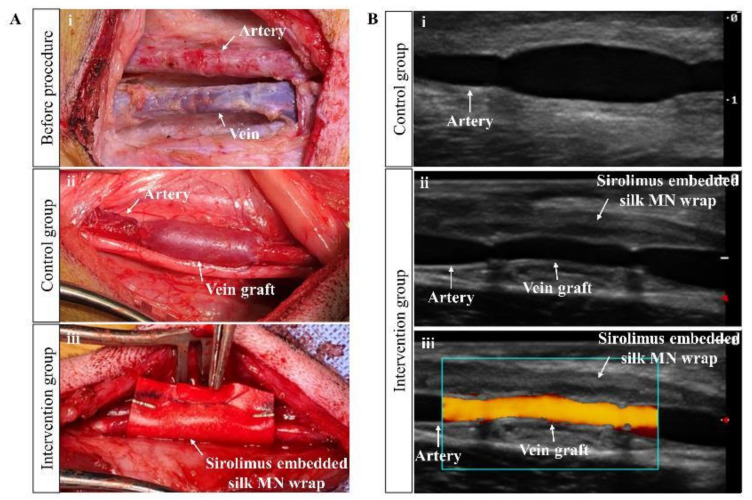
In vivo surgical procedures and ultrasonographic images. (**A**) Representative intra-operative images of (i) preimplantation, (ii) the control group, and (iii) the intervention group. (**B**) Representative postoperative ultrasonographic images in (i) the control group, (ii) the intervention group, and (iii) Doppler image of the vein graft in the intervention group. MN, microneedle.

**Figure 11 ijms-24-03306-f011:**
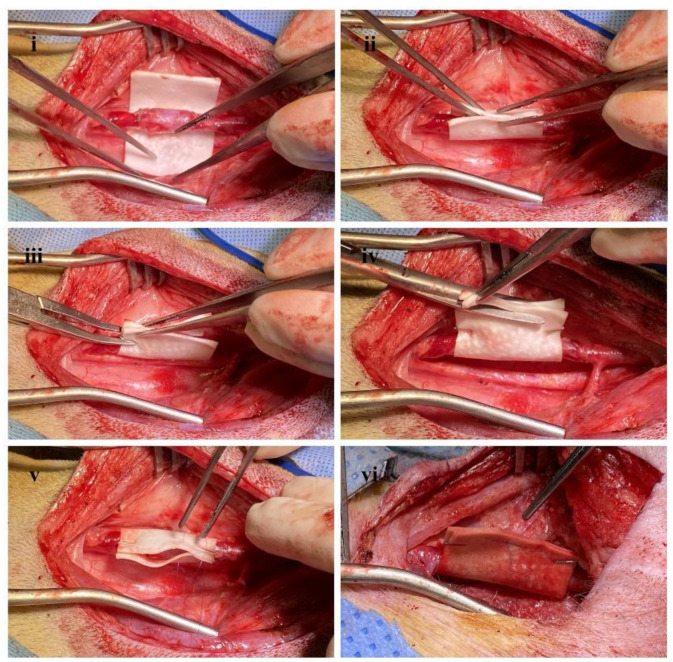
Detailed surgical procedures in the intervention group; (i) positioning of silk MN wrap to vein graft, (ii) wrapping the silk MN wrap around the vein graft, (iii) fixation of silk MN wrap with surgical clip, (iv) trimming of redundant wrap, (v) fixation of central portion with suture, (vi) completion of applying silk MN wrap to vein graft.

## Data Availability

Not applicable.

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
