# Peer review of "Sirolimus-Embedded Silk Microneedle Wrap to Prevent Neointimal Hyperplasia in Vein Graft Model"

_ijms, 2023, doi:10.3390/ijms24043306_

Round 1

Reviewer 1 Report

Microneedle wrap is an interesting drug delivery system used in different researches. 

This research used traditional used drug sirolimus and a silk microneedle wrap to inhibit neointimal hyperplasia in a dog vein graft model. 

The reviewer has several major concerns about the figures and control

1, Figure 5A, it does not make sense that no alpha-SMA expression in the inter group in femoral vein graft, since SMA is in the neointima and vein. PCNA in the pre jugular vein should be very low, but it showed much higher expression compared to the pre femoral; this does not make sense.  Figure 5 do need a double staining of PCNA and SMA. 

2, Figure 7, what is the rational to use cell autophagy?

3, For the proper control, the vein graft+blank silk MN wrap should be used.

4, A gross photograph of the interventional group should be added. 

5, The low power whole vein graft in all the groups should be added. 

Author Response

  1. Figure 5A, it does not make sense that no alpha-SMA expression in the inter group in femoral vein graft, since SMA is in the neointima and vein. PCNA in the pre jugular vein should be very low, but it showed much higher expression compared to the pre femoral; this does not make sense.  Figure 5 do need a double staining of PCNA and SMA. 

-Response

Thank you for your important comments. Intensity of protein expression in western blot expressed in relative amounts rather than absolute amounts. So, alpha-SMA expression in the intervention group seem to be shown no expression because difference in the quantitative amount of alpha-SMA between the intervention group and the pre/control group in femoral vein is much larger than in jugular vein. Similarly, PCNA expression of the pre group in jugular vein seem to be higher than femoral vein because difference in the quantitative amount of PCNA between the pre group and the control/intervention group in femoral vein is much larger than in jugular vein. Referencing the results in Figure 5B and 5C (quantitative analysis of alpha-SMA and PCNA) is thought to be helpful in interpreting the results.

 As your comments, double staining of PCNA and alpha-SMA should be very helpful interpreting the results. But it is hard to be performed because of the technical and timing problem. It needs additional protocol and reagents, but it takes enough time to prepare it. The purpose of measurement of alpha-SMA and PCNA was to investigate the mechanism of neointimal hyperplasia prevention because they indicated the smooth muscle cell proliferation and differentiation. I think that the purpose of the experiment can be reflected with the western blot itself.

  1. Figure 7, what is the rational to use cell autophagy?

-Response

Thank you for your insightful comments. Basal autophagy can protect plaque cells against oxidative stress by degrading damaged intracellular material, in particular polarized mitochondria. In this way, successful autophagy of the damaged components promotes cell survival [1]. The protective role of autophagy in atherosclerosis was illustrated by in vitro findings showing that SMC death induced by low concentrations of statins is attenuated by the autophagy inducer 7-ketocholesterol [2]. And induction of autophagy represents a vital component of a general stress response in vascular cells and could therefore be an important determinant of the stability of atherosclerotic plaques [3].

I added the discussion section as following:

“Several studies reported the beneficial role of autophagy in atherosclerosis. Kiffin et al. showed that autophagy of the damaged components could protect plaque cells against oxidative stress and could promote cell survival [46]. And Martinet et al. investigated the protective role of autophagy in atherosclerosis with in vitro experiment [47]. It showed that SMC death induced by low concentrations of statins was attenuated by the autophagy inducer 7-ketocholesterol. Schrijver et al. explained that inhibition of autophagy was harmful because this would accelerate other forms of cell death such as necrosis and im-pair the phagocytic process [48].”

[1] Kiffin R, Bandyopadhyay U, Cuervo AM. Oxidative stress and autophagy. Antioxid Redox Signal. 2006;8:152–162.

[2] Martinet W, Schrijvers DM, Timmermans JP, Bult H. Interactions between cell death induced by statins and 7-ketocholesterol in rabbit aorta smooth muscle cells. Br J Pharmacol. 2008;154:1236–1246.

[3] Schrijvers D. M., De Meyer G. R., Martinet W. Autophagy in atherosclerosis: a potential drug target for plaque stabilization. Arterioscler Thromb Vasc Biol 2011;31:2787–2791

.

  1. For the proper control, the vein graft+blank silk MN wrap should be used.

-Response

Thank you for your insightful comments. We performed unpublished preliminary study using bare (blank) silk MN wrap to the rabbit abdominal aorta injury model. This study showed bare silk MN wrap was not associated with neointimal hyperplasia prevention (below figure). So, we concluded that silk MN wrap itself without drug could not prevent neointimal hyperplasia and we did not include the bare silk MN wrap in this study.

We added comments at limitation section as following:

“We performed unpublished preliminary study using bare silk MN wrap without drug to the rabbit abdominal aorta injury model. This study showed bare silk MN wrap was not associated with prevention of neointimal hyperplasia (supplementary figure 1). So, we concluded that silk MN wrap itself without drug could not prevent neointimal hyperplasia and we did not include the bare silk MN wrap in this study.”

  1. A gross photograph of the interventional group should be added. 

-Response

 Thank you for your important comments. I added the figure 11 about the detailed surgical procedures in intervention group.

  1. The low power whole vein graft in all the groups should be added. 

-Response

 Thank you for your important comments. I understand the “low power” as “low pressure”. If my understanding is correct, the mean blood pressure was maintained at 60 mmHg during and after surgical procedures. The average low (diastolic) pressure was maintained at 45mmHg.

Reviewer 2 Report

The manuscript proposed by Jung-Hwan Kim et al., aims at evaluating sirolimus embedded silk microneedle wrap to prevent neointimal hyperplasia in vein graft model.

The manuscript is well written, clear, and convincing.

I have minor comments/remarks:

Line 79, “Figure 2B” correct is “Figure 1B”

Lines 276-287: the text should be in normal style and not in italics.

Figure 8 : the number of Tunel positive cells seems to be extremely high for a 200x magnification. Wouldn't there be a mistake ?

Author Response

  1. Line 79, “Figure 2B” correct is “Figure 1B”

-Response

Thank you for your important comments. I changed the “figure 2B” to “figure 1B”.

  1. Lines 276-287: the text should be in normal style and not in italics.

-Response

Thank you for your important comments. I change to normal style.

  1. Figure 8: the number of Tunel positive cells seems to be extremely high for a 200x magnification. Wouldn't there be a mistake?

-Response

 Thank you for your insightful comments. In others studies investigated the apoptotic cells using TUNEL staining, the TUNEL positive cell percentage in injury tissue was 10% in whole cells [1. 2]. Considering these findings, TUNEL positive cell counts in our study seem to be reasonable.

[1] Gurel Z, Zaro BW, Pratt MR, Sheibani N., Identification of O-GlcNAc modification targets in mouse retinal pericytes: implication of p53 in pathogenesis of diabetic retinopathy. PLoS One. 2014 May 1;9(5):e95561. doi: 10.1371/journal.pone.0095561.

[2] Ibrahim MM, et al., Myofibroblasts contribute to but are not necessary for wound contraction. Lab Invest. 2015 Dec;95(12):1429-38.

Round 2

Reviewer 1 Report

the author did not answer all the comments of this reviewer. 

The double staining is basic in research. 

This reviewer is still worry about the WB result. Same amount of housekeeping gene but with different target gene. Normal artery and vein are rarely proliferate, this result does not make sense. 

Author Response

  1. The double staining is basic in research.

-Response

Thank you for your important comments. As mentioned previous revision, double staining was hard to be performed because of the technical and timing problem. We add the comments in limitation section as following:

“Double staining of PCNA and α-SMA should be very helpful interpreting the results, but our study did not include these data.”

  1. This reviewer is still worry about the WB result. Same amount of housekeeping gene but with different target gene. Normal artery and vein are rarely proliferate, this result does not make sense.

-Response

Thank you for your insightful comments. Our previous study, there were numerus number of a-SMA in normal vein [1]. Normal cells also showed cell proliferation according to the cell cycle. Our study showed that sirolimus released by the MN wrap suppressed abnormal cell proliferation caused by the injury and prevented neointimal hyperplasia.
